# Microstructure and Mechanical Properties of Multicomponent Metal Ti(C,N)-Based Cermets

**Xuelei Wang** [1,*], **Qiufeng Wang** [2], **Zhaojun Dong** [3], **Xiaoqian Zhou** [1], **Xiaoliang Wang** [1], **Boyan Zhang** [1] **and Chao Meng** [1,*]

1   College of Materials Science and Engineering, Liaoning Technical University, Fuxin 123000, China; zxq6558960@126.com (X.Z.); ningke@163.com (X.W.); zhangboyan@163.com (B.Z.)
2   School of Geomatics, Liaoning Technical University, Fuxin 123000, China; wangqioufeng@163.com
3   College of New Energy and Environment, Jilin University, Changchun 130012, China; dongzhaojun@jlu.edu.cn
*   Correspondence: wangxuelei-19@163.com (X.W.); mengchao_ja@yeah.net (C.M.); Tel.: +86-0418-511-0099 (X.W.); +86-0418-511-0102 (C.M.)

**Abstract:** Ti(C,N)-based cermets with multicomponent ingredients were prepared using vacuum sintering technology. The effect of molding agents, binder phase and sintering temperature on Ti(C,N)-based cermets were studied. The optimum molding performance was obtained by adding 2% polyvinyl alcohol (PVA-1788). The microstructure and mechanical properties of Ti(C,N)-based cermets were investigated. The Ti(C,N)-based cermet with a weight percentage of TiC:TiN:Ni:Co:Mo:WC:Cr$_3$C$_2$:C = 40:10:20:10:7:8:4:1 and sintered at 1450 °C had the optimal mechanical properties. The relative bending strength, Vickers hardness, elastic modulus and wear resistance were 2010 MPa, 15.01 GPa, 483.57 GPa and 27 mg, respectively. Additionally, X-ray diffraction (XRD), backscatter scanning electron microscopy pictures (SEM–BSE), energy dispersive spectrometry (EDS) and optical micrographs of Ti(C,N)-based cermets were characterized.

**Keywords:** Ti(C,N)-based cermet; vacuum sintering; NiCoMo powder; microstructure; mechanical properties

## 1. Introduction

Ti(C,N)-based cermets are important wear-resistant structural materials [1]. They have a high temperature hardness, strength, good wear resistance, thermal conductivity and superior chemical stability [2–4]. However, compared with metal materials [5], the brittleness and impact toughness of this material still has obvious shortcomings that restrict its development for widespread application. In recent years, a lot of methods, such as spark plasma sintering technique (SPS) [6,7], hot-pressing sintering [8], high-frequency induction heated sintering method [9,10], liquid phase sintering [11] and the combination of powder metallurgy technique and colloidal method [12], etc., have been used to improve the fracture toughness of Ti(C,N)-based cermets. The SPS technique has the advantages of low cost, rapid response and high output. The processed products have the characteristics of high density and good uniformity by hot-pressing sintering. The liquid phase sintering speed is fast, the shrinkage is obvious, and the density can come close to the theoretical density. Research methods have improved the toughness but greatly decreased the strength and hardness. Therefore, improving the toughness of Ti(C,N)-based cermets without reducing their hardness and strength has become an urgent problem. The vacuum sintering method was applied in this study. Compared to other sintering methods, the vacuum sintering method had the following advantages: making the powder particles on the surface of the oxide reduction, increasing metal ceramic density, improving the combination of the binder and the hard phase, and achieving uniform phase distribution [13].

It is well known that the binder phase of metals and carbide has significant influence on the performance of Ti(C,N)-based cermets. In the binder phase of metals, Ni [14], Co [15], Mo [16] and Cr [17] metals are widely used. The addition of these metal binder phases can refine the grain size of matrix particles and produce the effect of fine grain strengthening, which improves the alloying strength of Ti(C,N)-based cermets. This results in an improvement of the alloying strength of Ti(C,N)-based cermets. Thus, the addition of various alloying elements to the matrix of the cermet is usually done by researchers to improve the performance of the cermet. In addition, carbide, such as WC [18,19], $Cr_3C_2$ [20], VC [21], HfC [22], TaC [23] and ZrC [24], is often used to modify the mechanical properties of Ti(C,N)-based cermets. However, the effect of combining two carbides on the mechanical properties of Ti(C,N)-based cermets is rarely reported and studied. The function of the C element in Ti(C,N)-based cermets is to adjust the thickness of the outer ring phase of the core–ring structure and the volume fraction of the binder phase [25]. For example, Sun et al. [26] investigated the effect of short carbon fiber concentration on the mechanical properties of Ti(C,N)-based cermets and stated that the bending strength was 643.3 MPa and the hardness was 79.1 HRA. In addition, the binder has an important effect on molding properties, but there is less related research. On the other hand, the sintering temperature is also an important parameter to the performance.

Therefore, this paper studied TiC and TiN as the hard phase; Ni, Co and Mo as the binders; and WC and $Cr_3C_2$ as the carbide additives. Subsequently, a small amount of C was added. Finally, Ti(C,N)-based cermets were synthesized under the condition of eight components, and the influence of metal binders and sintering temperature on properties for Ti(C,N)-based cermets was studied.

## 2. Experimental Section

### 2.1. Materials

The specifications and manufacturers of raw powders are shown in Table 1. The basic information about molding agents is shown in Table 2. The SEM morphology of various powder samples is shown in Figure 1.

**Table 1.** Basic information about raw powders.

| Powders | Purity (%) | Size (μm) | Manufacturer |
| --- | --- | --- | --- |
| TiC | ≥99.0 | 2.40 | Qinghe bodrill metal material Co., Ltd., Xingtai, China |
| TiN | ≥99.0 | 2.50 | Qinghe bodrill metal material Co., Ltd., Xingtai, China |
| Ni | ≥99.7 | 2.25 | Shanghai Yunfu Nanotechnology Co., Ltd., Shanghai, China |
| Co | ≥99.0 | 3.60 | Shanghai Yunfu Nanotechnology Co., Ltd., Shanghai, China |
| Mo | ≥99.9 | 2.30 | Shanghai Yunfu Nanotechnology Co., Ltd., Shanghai, China |
| WC | ≥99.9 | 3.40 | Qinghe bodrill metal material Co., Ltd., Xingtai, China |
| $Cr_3C_2$ | ≥99.0 | 2.30 | Qinghe bodrill metal material Co., Ltd., Xingtai, China |
| C | ≥99.8 | 3.50 | Qinghe bodrill metal material Co., Ltd., Xingtai, China |

**Table 2.** Basic information about molding agents.

| Samples | Density (g/cm³) | Manufacturer |
| --- | --- | --- |
| Paraffin wax | 0.82 | Aladdin Chemistry Co., Ltd., Shanghai, China |
| Polyethylene glycol-2000 (PEG-2000) | 1.13 | Aladdin Chemistry Co., Ltd., Shanghai, China |
| Polyvinyl alcohol-1788 (PVA-1788) | 1.30 | Aladdin Chemistry Co., Ltd., Shanghai, China |

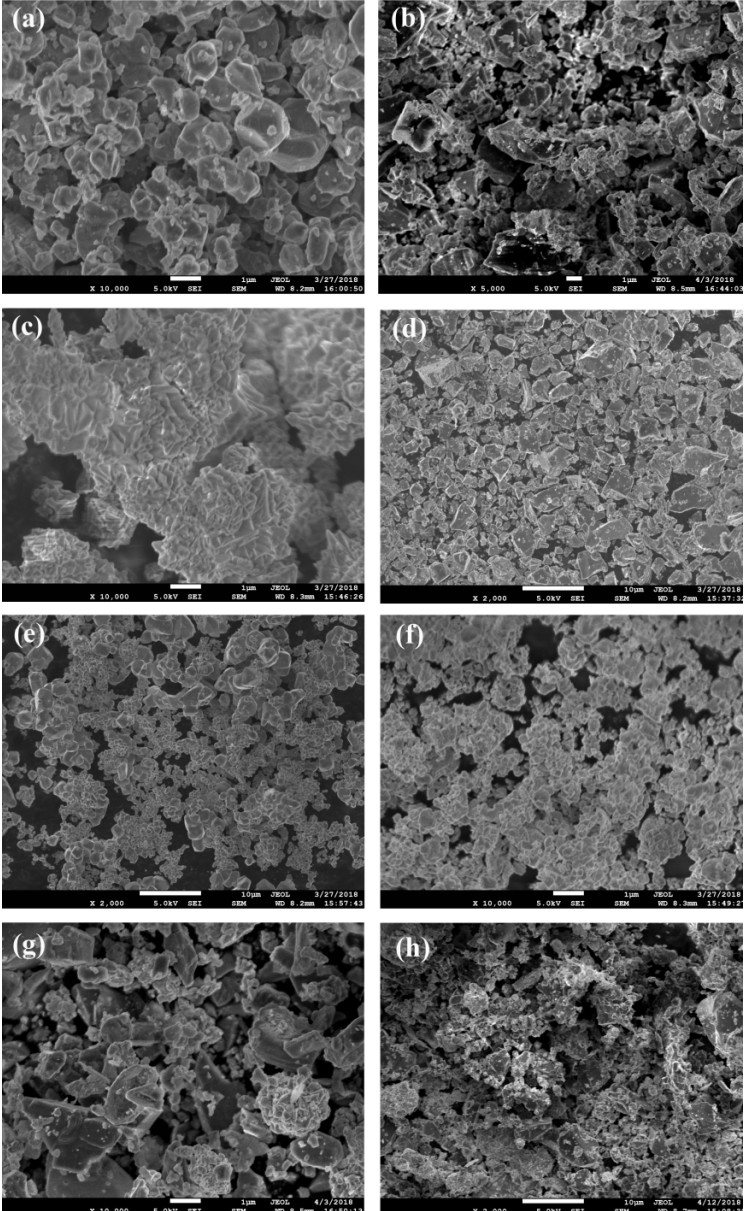

**Figure 1.** SEM of powders: (**a**) TiC; (**b**) TiN; (**c**) Ni; (**d**) Co; (**e**) Mo; (**f**) WC; (**j**) $Cr_3C_2$; (**h**) C.

## 2.2. Synthesis

Ti(C,N)-based cermets were prepared using the vacuum sintering method. The proportion of experimental powers is shown in Table 3. The sintering temperature of the Ti(C,N)-based cermets is shown in Table 4. According to the experimental design, the corresponding raw material powder was weighed and mixed for ball mill. Anhydrous ethanol was added into the ball mill tank. The WC ball was used as a ball mill medium. The ball material ratio was 7:1, and the ball mill time was 24 h. The speed of the planetary ball mill was 300 rpm. After ball grinding, the powder was dried in a vacuum at 80 °C. In this experiment, polyvinyl alcohol, which was added at 2 wt.% of the powder mass, was used as the molding agent. The powders were ground and mixed well. The powders were compacted by 50 KN to press them into a Φ 50 mm green compact, using a holding time set at 5 min. The de-binding and sintering process curve of Ti(C,N)-based cermets are shown in Figure 2a,b (represented by 1450 °C). Other sintering temperatures were 1350 °C, 1400 °C and 1500 °C. The compacted embryo was sintered in a vacuum (~$1.0 \times 10^3$ Pa) sintering furnace.

**Table 3.** Composition design of Ti(C,N)-based cermets (wt.%).

| Samples | TiC | TiN | Ni | Co | Mo | WC | Cr$_3$C$_2$ | C |
|---------|------|-----|----|------|----|----|------|---|
| 10Ni | 55 | 10 | 10 | 5 | 7 | 8 | 4 | 1 |
| 20Ni | 40 | 10 | 20 | 10 | 7 | 8 | 4 | 1 |
| 25Ni | 32.5 | 10 | 25 | 12.5 | 7 | 8 | 4 | 1 |
| 30Ni | 25 | 10 | 30 | 15 | 7 | 8 | 4 | 1 |

**Table 4.** Different sintering temperatures of Ti(C,N)-based cermets.

| Name | 1350 °C | 1400 °C | 1450 °C | 1500 °C |
|------|---------|---------|---------|---------|
| 10Ni | 10Ni-1350 °C | 10Ni-1400 °C | 10Ni-1450 °C | 10Ni-1500 °C |
| 20Ni | 20Ni-1350 °C | 20Ni-1400 °C | 20Ni-1450 °C | 20Ni-1500 °C |
| 25Ni | 25Ni-1350 °C | 25Ni-1400 °C | 25Ni-1450 °C | 25Ni-1500 °C |
| 30Ni | 30Ni-1350 °C | 30Ni-1400 °C | 30Ni-1450 °C | 30Ni-1500 °C |

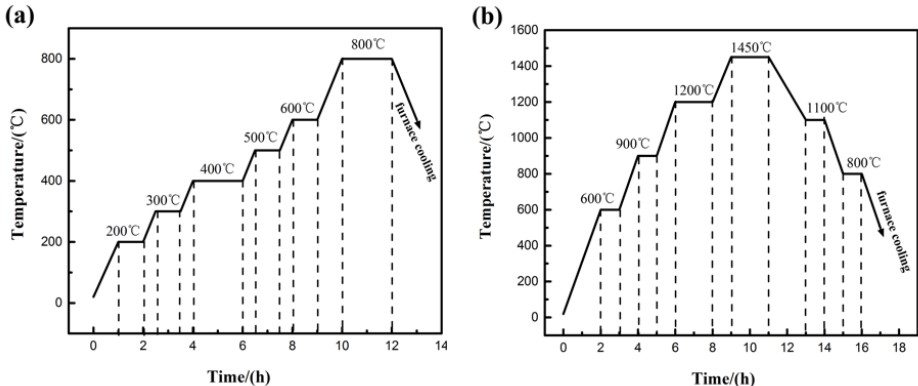

**Figure 2.** (**a**) De-binding curve and (**b**) sintering process curve of Ti(C,N)-based cermets.

## 2.3. Mechanical Properties Test

Rockwell hardness (HRSS-150, Shanghai Caikang Optical Instrument Co., Ltd., Shanghai, China) was used to measure the hardness of Ti(C,N)-based cermets. Three replicates were carried out for all the investigations in the present study.

The abrasive wear testing machine (ML-100, Jinan Evergrande Huifeng Test Instrument Co., Ltd., Jinan, China) was used to evaluate the abrasive wear resistance of the material. P320 abrasive paper was selected as the friction pair, loading was 20 N and wear time was 10 min. An electronic weighing balance with an accuracy level of 0.1 mg was used to measure the weight loss of the sample.

A three-point bending experiment was carried out by a universal testing machine (WAW-100E, Jinan New Gold Testing Machine Co., Ltd., Jinan, China). The sintered sample was cut into testing specimens by the electrical discharge wire cutting method, and the dimension of the specimen was 40 mm × 4 mm × 3 mm. The formula for calculating the bending strength at three points was:

$$\sigma_b = \frac{3PL}{2bh^2} \tag{1}$$

$$E_b = \frac{L^3}{4bh^3} \times \frac{\Delta P}{\Delta d} \tag{2}$$

$\sigma_b$ is the bending strength (MPa), $E_b$ is elastic modulus (GPa), $P$ is the maximum fracture load (N), $L$ is the span between two points (mm), $b$ is the width of the sample cross section (mm), $h$ is the height of the sample cross section (mm) and $\Delta P/\Delta d$ is the slope of the stress–strain curve.

### 2.4. Microstructure Observation

The composition of Ti(C,N)-based cermets was measured by an X-ray diffraction (λ = 1.5418 Å) device (Rigaku 6100, Rigaku Inc., Tokyo, Japan). The surface morphologies of the samples were characterized by using an optical microscope (Axiovert-40-MAT, ZEEISS Inc., Oberkochen, Germany). A scanning electron microscope (JSM7500F, JEOL Inc., Tokyo, Japan) equipped with energy dispersive spectrometry (EDS) (X-max, Oxford Instruments, Oxford, UK) and backscatter scanning electron microscopy pictures (SEM–BSE) were used to analyze the element and phase distribution.

## 3. Results and Discussion

### 3.1. Synthesis Process

TiC and TiN powders were used as the hard phase, while Ni, Co and Mo powers were believed to act as the binder phase, and WC, $Cr_3C_2$ and C played modified role. It was difficult to de-bind the mixed powder due to the many powder samples being added. In order to solve this problem, the influence of paraffin wax, PEG-2000 and PVA-1788 molding agents on de-binding was explored. As shown in Figure 3a, the paraffin as a molding agent showed poor formability. In the de-binding process, cermets were most likely to loosen or be crushed. When PEG-2000 was used as a molding agent, the cermets were layered and stripped, as shown in Figure 3b. The PVA-1788 as a forming agent had the best forming capacity among the three molding agents, as shown in Figure 3c. Finally, this experiment selected PVA-1788 as the forming agent for preparation of Ti(C,N)-based cermets. The decomposition temperature of PVA-1788 was 400 °C. A longer holding time was set at 400 °C for completely removing the PVA-1788 in the de-binding process (Figure 2a). A longer preservation time was set at 800 °C to ensure that the degreased embryo body had a certain strength, so as to prevent the sinter body from collapsing during the sintering process. In order to effectively eliminate the gas generated in the sintering process, 1200 °C was set for the insulation steps (Figure 2b). The reason for setting the heat preservation step at 1100 °C after sintering is that it can prevent cracks or bending caused by excessive cooling during the cooling process [27].

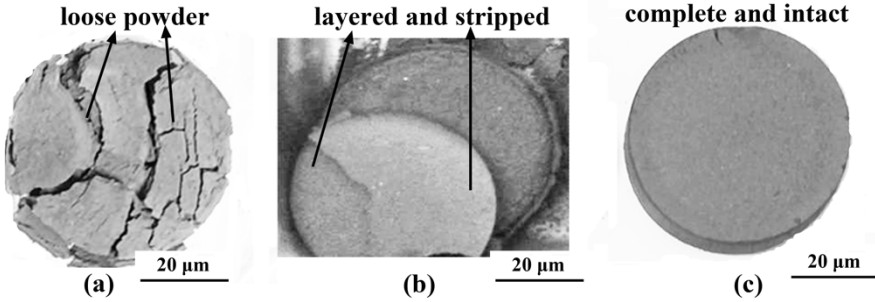

**Figure 3.** Formability of different molding agents: (**a**) paraffin; (**b**) PEG-2000; (**c**) PVA-1788.

### 3.2. XRD Phase Analysis

As shown in Figure 4, four samples of Ti(C,N)-based cermets—20Ni-1350 °C, 20Ni-1450 °C, 25Ni-1350 °C and 25Ni-1450 °C were selected as representatives for the XRD phase analysis. Compared to the a and c, or b and d samples, it was found that when the ratio was consistent with the higher temperatures of the c and d samples, the Ni metal binder phase and carbide morphology characteristics weakened gradually. The increase of temperature led to the coarsening of the microstructure of the ceramic core phase, which inhibited the growth process of the Ni phase recrystallization. TiC, $Cr_3C_2$, WC and $Mo_2C$ carbides [28–30] formed the corresponding metal solid solution. As Mo and W elements with a larger atomic radius were dissolved in the Ti(C,N)-based cermets, the lattice spacing of the Ti(C,N)-based cermets increased, and the characteristic diffraction peak shifted to the left (the position of the three dotted lines). Compared to the a and b, or c and d samples, when the sintering

temperature was consistent, the higher binding phase of the Ti(C,N)-based cermets showed higher characteristics of diffraction peak. At the same sintering temperature, the high bond phase was conducive to crystallization.

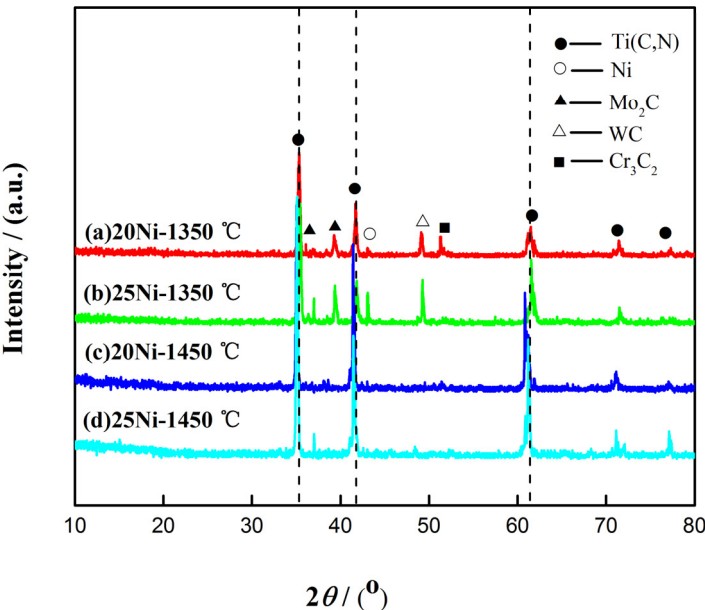

**Figure 4.** XRD of Ti(C,N)-based cermets: (**a**) 20Ni-1350 °C; (**b**) 25 Ni-1350 °C; (**c**) 20Ni-1450 °C; (**d**) 25 Ni-1450 °C.

### 3.3. Microstructure and Composition of the Cermets

Metallographic porosity was measured for samples with different binder content at 1400 °C (Figure 5). According to international standards (ISO/BS4505), type A has an aperture less than 10 μm and usually contains A02 (0.02%), A04 (0.06%), A06 (0.2%) and A08 (0.6%). Type B has an aperture of 10–25 μm and contains B02 (0.02%), B04 (0.06%), B06 (0.2%) and B08 (0.6%). The test results are shown in Table 5. With the increase of binder amount, the porosity of the material decreased gradually. When the content of the binder phase was sufficient, the wettability at the hard phase was good. The capillary forces in the pores and the viscous flow dynamics of the bonder phase made the position of the hard phase particles move and rearrange, thus rapidly reducing the porosity of the material. Therefore, with the increase of the bonder phase addition in the material, the dissolution and precipitation processes were sufficient, leading to a decrease in the porosity of the material. This indicates that the addition of the bonder phase has a great influence on the porosity of the final sintered body.

The SEM–BSE micrographs of the different sintering temperature are exhibited in Figure 6. The phenomenon of hard phase particle aggregation is obvious at the low sintering temperature (Figure 6a). At the sintering temperature of 1400 °C, the ring phase is thin and no core structure is greater (Figure 6b). This non-uniform microstructure was due to an inadequate alloying degree at a relatively low sintering temperature. With further increasing of the sintering temperatures, the microstructure became more uniform, the thickness of the ring phase increased and the hard phase particles became more refined (Figure 6c). This is because with the increase of the sintering temperature, the diffusion rate was accelerated, the dissolution rate of the hard bonder phase was improved and the dissolution–precipitation process was promoted. Therefore, each component in the binder phase tended to be more homogeneous through diffusion, and the alloying degree was greatly increased with the sintering temperature at 1450 °C. When the sintering temperature was further increased, the hard phase decreased and the ring phase thickened (Figure 6d). The ring phase is the brittle phase. Excessive thickness will damage the properties of the cermets [31].

**Table 5.** Effect of the bonder phase addition on the porosities of Ti(C,N)-based cermets.

| Cermets | 15Ni-1450 °C | 20Ni-1450 °C | 25Ni-1450 °C | 30Ni-1450 °C |
|---------|--------------|--------------|--------------|--------------|
| Porosity | A08B06 | A06B06 | A04B04 | A02B02 |

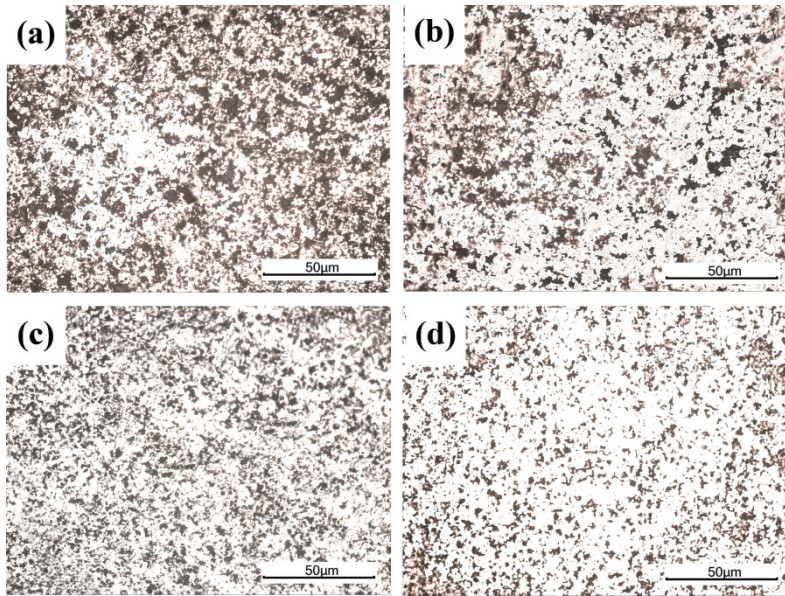

**Figure 5.** Light-optical microscopy of Ti(C,N)-based cermets: (**a**) 15Ni-1450 °C; (**b**) 20Ni-1450 °C; (**c**) 25Ni-1450 °C; (**d**) 30Ni-1450 °C.

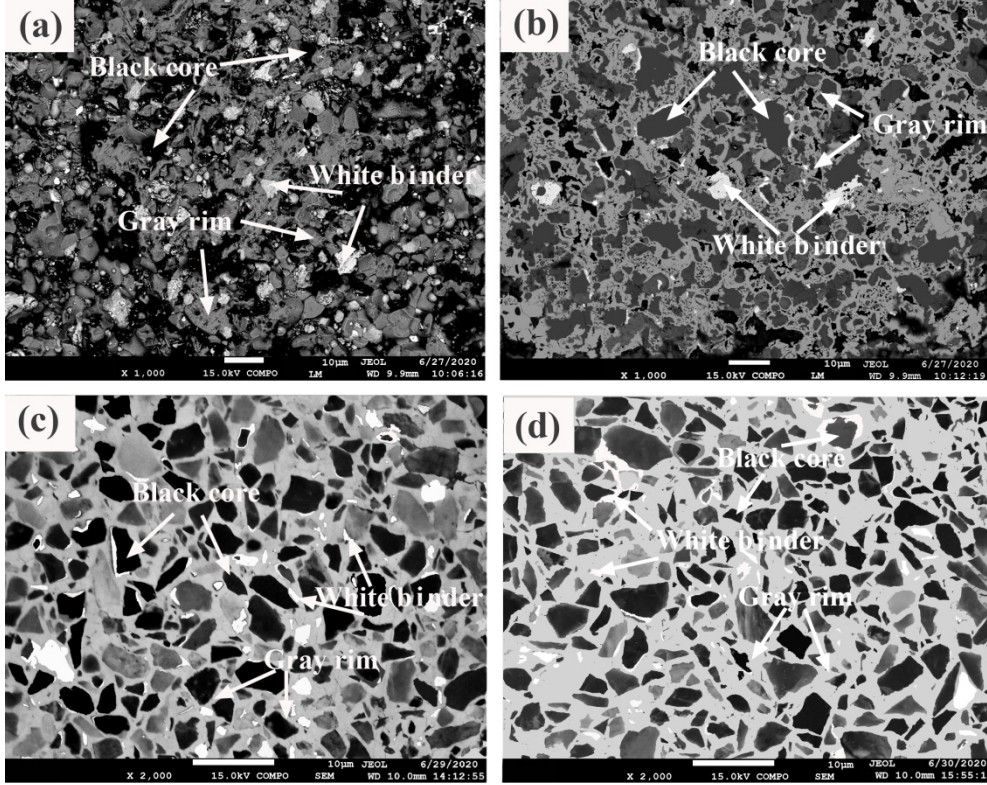

**Figure 6.** SEM–BSE micrographs for Ti(C,N)-based cermets: (**a**) 20Ni-1350 °C; (**b**) 20Ni-1400 °C; (**c**) 20Ni-1450 °C; (**d**) 20Ni-1500 °C.

The elemental mapping of the 15Ni-1450 °C sample is shown in Figure 7. According to the elemental mapping, the black core is the ceramic hard phase, which is dominated by Ti(C,N); the gray rim is the ring phase, which is dominated by Ti, Ni, Co, Mo carbides and carbonitride; and the white bonder is the binder phase, which is dominated by metal alloys with the large atomic numbers of Ni, Co, Mo and W elements. EDS point scanning analysis was tested to determine the element content in each phase (Figure 8). The specific values are shown in Table 6. Since the positions of N and C coincide, it can be inferred that the black core is mainly composed of Ti, C and N elements. The gray shell is the intermediate transition phase, which is mainly composed of a (Ti,W,Mo)(C,N) solid solution. Ti, W and other metal elements were dissolved in the NiCoMo bond phase, and these elements played the role of the solid solution strengthening bond phase.

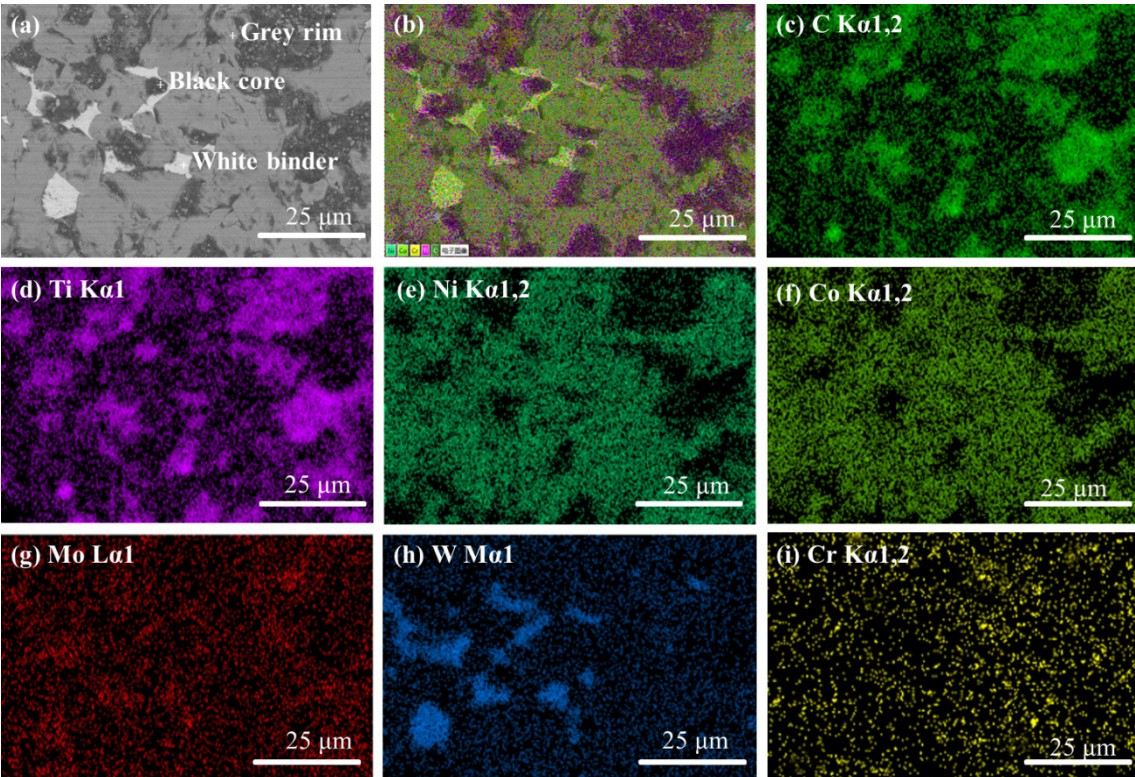

**Figure 7.** SEM–BSE micrographs and elemental mapping of Ti(C,N) ceramics.

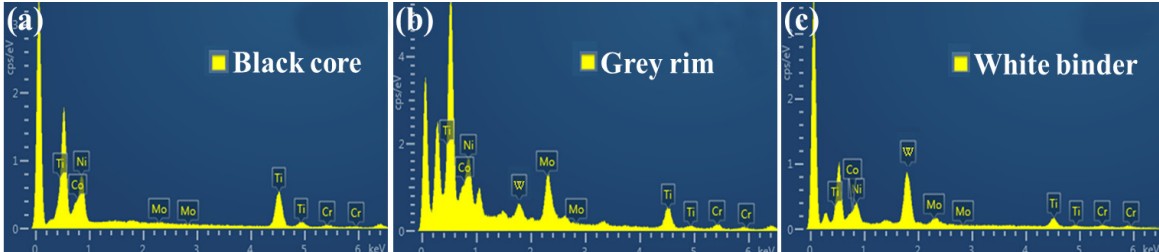

**Figure 8.** EDS point scanning results: (**a**) black core; (**b**) grey rim; (**c**) white binder.

**Table 6.** EDS of Ti(C,N)-based cermets (wt.%).

| Element | Ti | C | Ni | Co | Mo | Cr | W | Total |
|---|---|---|---|---|---|---|---|---|
| Black core | 48.22 | 33.87 | 7.68 | 2.82 | 1.67 | 2.11 | 3.63 | 100 |
| Grey rim | 33.68 | 22.86 | 15.37 | 13.56 | 1.93 | 8.76 | 3.84 | 100 |
| White binder | 11.07 | 7.15 | 22.24 | 19.81 | 3.85 | 1.56 | 33.32 | 100 |

### 3.4. Mechanical Properties of the Ti(C,N)-Based Cermets

The influence of binder phase content and sintering temperature for the mechanical properties of Ti(C,N)-based cermets was analyzed. With the increase of sintering temperature, the hardness of the 30Ni and 25Ni samples with a high binder content decreased, while the hardness of the 20Ni and 15Ni samples with a low binder content first increased and then decreased (Figure 9a). In the Ti(C,N) cermet with a relatively high binder content, alloying could be realized when the sintering temperature was low. When the sintering temperature continued to increase, the TiN decomposed, the hard phase and binder phase stratified and the microstructure was not uniform, which led to a lower hardness [32]. At the same temperature, the higher the content of the binder phase, the lower the content of the hard phase, so the hardness was lower. With increase of the sintering temperature, the bending strength first increased and then decreased (Figure 9b). The temperature corresponding to the peak of bending strength gradually decreased with the increase of the content of the binder phase. This is because with the increase of sintering temperature, the volume of the liquid phase increased. This was conducive to particle rearrangement, making the solid–liquid interface more stable and reducing the porosity of the material while improving the compactness of the material. However, when the sintering temperature was too high, the TiN underwent a decomposition reaction and reduced the density. The elastic modulus and bending strength had similar behaviors with the changes of sintering temperature and the content of bond phase. The difference is that the value of the elastic modulus was not only related to the bending strength, but also affected by its hardness. By comparing Figure 9a,d, the variation trend was the opposite. This is because hardness was the decisive factor affecting wear resistance, excluding other factors [33]. The above data is the average of six measurements, and the average error is within 5%.

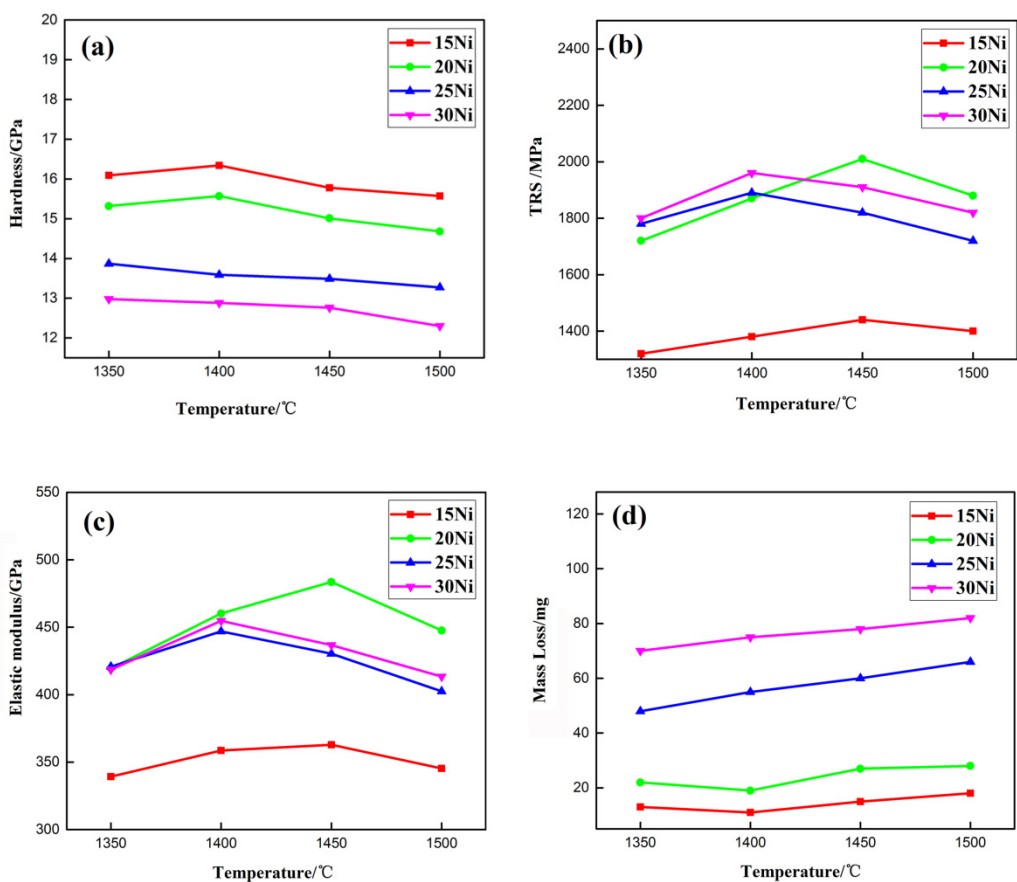

**Figure 9.** Effect of metal binder phase content and sintering temperature on Ti(C,N)-based cermets: (**a**) Vickers hardness; (**b**) TRS; (**c**) elastic modulus; (**d**) wear resistance.

According to the above analysis, it was found that the sample with the greatest hardness and wear resistance was 15Ni-1400 °C, with the hardness reaching 16.34 GPa. The sample with the highest bending strength was 20Ni-1450 °C, with the bending strength reaching 2010 MPa. Its hardness value was 15.01 GPa, and its wear resistance performance was also good. However, 15Ni-1400 °C showed poor bending strength and density, with a bending strength of 1380 MPa. Accordingly, it can be concluded that 20Ni-1450 °C has the best comprehensive mechanical properties. Comparing other studies: Borrell et al. [34] processed a TiCN-Co cermet, which had a Vickers hardness of 17.1 GPa and a bending strength of 904 MPa; Alvarez et al. [35] prepared a Ti(CN)-based cermet with a TiCN-$Mo_2C$-Ni component, which exhibited a Vickers hardness of 18 GPa; Yin et al. [36] investigated the effect of a WC/$Mo_2C$ ratio on the mechanical properties of a Ti(CN)-based cermet and reported that the hardness was 16.83 GPa; Verma et al. [6] investigated a TiCN-WC-Ni/Co Ti(CN)-based cermet showing a Vickers hardness of 16.01 GPa; Wu et al. [37] prepared a Ti(CN)-based cermet with a TiC-TiN-WC-Mo-Ni-$Cr_3C_2$-C component, which gave a bending strength of 1900 MPa and 90.6 HRA. According to the above results, it can be concluded that the prepared cermets were deficient in hardness, but had great advantages in bending strength.

## 4. Conclusions

Ti(C,N)-based cermets with multicomponent ingredients were prepared using vacuum sintering technology. The microstructure and mechanical properties of Ti(C,N)-based cermets were observed and tested by various performance testing methods. The following conclusions were obtained from this study:

(1) The role of paraffin wax, PEG-2000 and PVA-1788 molding agents on Ti(C,N)-based cermets was explored. When the amount of PVA-1788 was added at 2 wt.%, the sintering result of the Ti(C,N)-based cermets was optimal.

(2) The 20Ni-1450 °C sample had the best comprehensive mechanical properties: its bending strength was 2010 MPa, its hardness value was 15.01 GPa and its elastic modulus was 483.57 GPa.

**Author Contributions:** Data curation, Q.W. and B.Z.; formal analysis, X.W. (Xuelei Wang) and Z.D.; funding acquisition, X.W. (Xiaoliang Wang) and C.M.; methodology, X.Z.; software, B.Z.; supervision, Q.W.; writing—original draft, X.W. (Xuelei Wang); writing—review and editing, C.M. All authors have read and agreed to the published version of the manuscript.

**Funding:** This study was supported by the National Natural Science Foundation of China (51974152 and 51805235), the Joint Research Fund Liaoning-Shenyang National Laboratory for Materials Science (2019JH3/30100025) and College Students' innovation training program (201910147119).

**Conflicts of Interest:** The authors declare no conflict of interest.

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
