# Peer review of "Microstructure and Mechanical Properties of Multicomponent Metal Ti(C,N)-Based Cermets"

_metals, doi:10.3390/met10070927_

Round 1

Reviewer 1 Report

The paper deals with Ti(C,N) cermets which are bonded with a Ni-Co alloy.

The following items require a substantial rewriting of the paper, and perhaps some re-investigation:

Table 1: What is size of powders? By which measurement? By FSSS, by BET?

Figure 1: Not necessary in scientific journal. “Metals” is not a textbook for students.

Figure 3: It is clear and well-known to the scientific community. Not necessary.

Figure 5: A TiCo phase should not form, there is TiC and TiN in the starting formulation, these do not react to TiCo. It is also questionable why free Mo2C, WC and Cr3C2 are identified. These phases form a complex (Ti,W,Mo)(C,N) phase in sintered cermets and Cr3C2 dissolves in the binder (see Sanchez, Lengauer et al.).

Figure 8: Data are very much affected by the poor lateral resolution of SEM ( e.g. slight shift in location change the data substantially and surrounding phases influence the measurements). The figures have therefore no substantial meaning.

Figure 9 and 10 do not present significant data. It is important to define porosity (e.g. ISO) and to show micrographs to proof the quality of the samples (porosity).  

Hardness should be given in Vickers (HV) for comparison with literature. A deep comparison with known data (see literature) is not provided.

Literature: Obviously important and standard literature was not consulted for getting deep insight of what other researchers gained. See e.g. papers of the cermet group in Vienna (Lengauer et al.), many papers on cermets with a very similar type (hard phase, binder phase, Cr-addition) were published by this group (see Journals: Int. J Refractory metals and hard materials, Metals, Solid State Phenomena, see also ResearchGate). TiCN-Fe like refs. 3, 16 is on the other hand not relevant because of the different binder phase.

Reviewer 2 Report

Please see attached review.

Reviewer 3 Report

A worthwhile study exploring some aspects of structure-property relationships in Ti(C,N)-based cermets and the effect of binder and addition chemistry.

There is room for revision in terms that the introduction could be strengthened by referring to some aspects of WC-based cermets in terms of binder and carbide effects and their structure-property relation. For instance, studies of fractography and binder phase chemistry from Roebuck and Mingard in WC-binder materials.

There is also the comment about vacuum sintering being selected as the best sintering method but no other sintering methods were mentioned.

More expansion on carbon and its effects on Ti(C,N) morphology and composition would be relevant. More needs to be made of the de-binding section as this is an interesting part of the paper and is often neglected in other studies.

The nomenclature needs revision - the paper in its present form uses non-standard vocabulary ('powdered compact' instead of 'embryo' and 'de-binding' not 'degreasing'). Revision of this will go some way to improve clarity of the paper. This is particularly the case for the de-binding study.

Round 2

Reviewer 1 Report

Manuscript seems amended well now, but please submit a version without the correction marks. It is not easily possible to read it with correction marks and finally you have to supply a version without these marks.

Reviewer 2 Report

  • Manuscript contains some possible English translation errors. Still contains minor English grammar errors.
  • The explanation of keeping the Ni/Co ratio fixed at 2 resolves the concern around the experimental design. Thank you for the detailed explanation.
  • The response to the binder usage is acceptable. I agree that this is not always discussed in sufficient detail given how much it can affect the process of the powder compact.
  • The added figures are beneficial to the discussion, and the microstructure images are much improved. However, Figure 5 is difficult to see in the manuscript and may need further brightness and contrast adjustment.
  • Discussion is improved. The results are more quantitative. The conclusions are more clearly supported by the presented results.
  • Ln 210 to 211: this statement needs a supporting reference.
  • Ln 277 to 279: this statement needs a supporting reference
  • Added discussion in Section 3.4 needs an appropriate reference for the
  • Reviewer comments have been adequately addressed.
  • Recommendation is for acceptance with minor revision.
